# Culture Supernatant of *Enterococcus faecalis* Promotes the Hyphal Morphogenesis and Biofilm Formation of *Candida albicans*

**DOI:** 10.3390/pathogens11101177

**Published:** 2022-10-12

**Authors:** Qingsong Jiang, Qi Jing, Biao Ren, Lei Cheng, Xuedong Zhou, Wenli Lai, Jinzhi He, Mingyun Li

**Affiliations:** 1State Key Laboratory of Oral Diseases, National Clinical Research Center for Oral Diseases, West China School of Stomatology, Sichuan University, Chengdu 610041, China; 2Department of Orthodontics, West China School of Stomatology, Sichuan University, Chengdu 610041, China; 3Department of Cariology and Endodontics, West China School of Stomatology, Sichuan University, Chengdu 610041, China

**Keywords:** *Enterococcus faecalis*, *Candida albicans*, interaction, supernatant, biofilm

## Abstract

(1) Background: *Enterococcus faecalis* and *Candida albicans* are often isolated from infected root canals. The interaction between these two species is not clear enough. Therefore, the objective of this study was to investigate the effect of *E. faecalis* on the biofilm formation, hyphal morphogenesis and virulence gene expression of *C. albicans*. (2) Methods: We used the culture supernatant of *E. faecalis* (CSE) to treat the biofilms of *C. albicans*. Then, crystal violet staining and colony-forming unit (CFU) counting were performed to evaluate biofilm biomass. Scanning electron microscopy (SEM) and confocal laser scanning microscope (CLSM) were applied to observe fungal morphology. Subsequently, exopolymeric substances (EPS) production, cellular surface hydrophobicity (CSH) and adhesion force of biofilms were investigated by CLSM, water–hydrocarbon two-phase assay and atomic force microscopy (AFM), respectively. Finally, the expression of *C. albicans* virulence genes (*ALS1*, *ALS3*, *HWP1* and *EFG1*) were measured by RT-qPCR assay. (3) Results: The exposure of CSE promoted the biofilm formation and hyphal morphogenesis of *C. albicans*, increased the EPS production, CSH and adhesion force of *C. albicans* biofilms, and increased the expression level of *EFG1*. (4) Conclusions: Our data indicated that CSE promoted the hyphal morphogenesis and biofilm formation of *C. albicans.*

## 1. Introduction

*Enterococcus faecalis* is a common species in secondary infected root canals and associated with the failure of endodontic treatment [1,2]. The detection rate of *E. faecalis* in failed root canal therapy varies from 12% to 90%, depending on different methodologies [3,4]. *In vitro* studies demonstrated that *E. faecalis* could penetrate into the dentin tubules up to 100 μm, and *E. faecalis* inoculated into root canals could be revived after 12 months, making the elimination difficult [5,6]. *Candida albicans* was the most frequently isolated fungus in infected root canals [7]. In addition to root canals, *E. faecalis* and *C. albicans* co-exist in the gastrointestinal tract and urogenital tract [8,9]. *C. albicans* can form a thick biofilm structure, which is more than 200 μm in depth, to resist antifungal treatment and to evade the immune system. The biofilm formation process of *C. albicans* contains four major steps: adherence, proliferation, maturation and dispersal [10]. *C. albicans* mature biofilms contain yeast and hyphal cells which are encased in exopolymeric substances (EPS). The hyphal cells form a complicated network to increase the robustness of the biofilm [11].

There are increasing studies focusing on the interaction between *C. albicans* and bacteria. Many bacterial species have been found to exhibit a synergic relationship with *C. albicans*, including *Staphylococcus aureus*, *Streptococcus mutans*, *Streptococcus gordonii* and *Bacteroides fragilis* [12,13,14,15,16]. However, the interaction between *E. faecalis* and *C. albicans* is still controversial. On the one hand, a bacteriocin from *E. faecalis*, EntV, was reported to suppress the hyphal formation, biofilm formation and virulence of *C. albicans* [17]. On the other hand, *E. faecalis* was found to increase the tissue invasion of *C. albicans* with an organotypic oral epithelial model [18]. Therefore, the cross-kingdom interaction between these two species is more complicated than what we understand, and more studies are required to better clarify it. The objective of this study was to analyze the effect of *E. faecalis* on the biofilm formation of *C. albicans* in vitro, providing more evidence to explain the interaction between these two microorganisms.

## 2. Results

### 2.1. Culture Supernatant of E. faecalis (CSE) Could Promote the Biofilm Formation of C. albicans

We measured the effect of culture supernatant of *E. faecalis* (CSE) and boiled CSE on *C. albicans* biofilm formation by crystal violet staining. As shown in Figure 1A, the CSE and the boiled CSE treated groups showed more biofilm biomass than the control group. The stereomicroscope images also showed thicker biofilms in the CSE and the boiled CSE treated groups (Figure 1B). To quantify the live cells in *C. albicans* biofilms, we applied a colony-forming unit (CFU) counting assay. The data showed that both CSE and boiled CSE increased the CFUs in biofilms significantly (Figure 1C), which was consistent with the results of crystal violet staining. Then we investigated *E. faecalis* cells and intracellular content on the biofilm formation of *C. albicans*. Interestingly, both the cells (Appendix A) and the intracellular content (Appendix A) of *E. faecalis* exhibited no effect on *C. albicans* biofilms.

### 2.2. CSE Promoted Hyphal Morphogenesis of C. albicans

Given that *C. albicans* biofilms contain yeast and hyphal cells, hyphal morphogenesis could affect the biofilm biomass. To observe the morphology of *C. albicans*, we performed Scanning electron microscopy (SEM) and Confocal laser scanning microscopy (CLSM) assays. The images showed that CSE promoted hyphal formation of *C. albicans* (Figure 2A,B).

### 2.3. CSE Increased the EPS Production of C. albicans Biofilms

To investigate the EPS production of *C. albicans* biofilm, we used SYTO 9 to label cells (green) and Alexa Fluor 647 to label exopolymeric substances (EPS) (red). The CLSM images showed that there were more hyphae and more EPS in the CSE treated biofilms (Figure 3A). Meanwhile, the statistical analysis revealed that CSE increased the EPS/cell ratio in biofilms. (Figure 3B)

### 2.4. CSE Altered CSH of C. albicans

Adherence and morphogenesis are two crucial stages in the *C. albicans* biofilm development process [19], the first of which is positively associated with cell surface hydrophobicity (CSH) [20]. Therefore, we performed a water–hydrocarbon two-phase assay to evaluate the CSH of *C. albicans*. We found the CSH to be 0.147 in the control group and 0.319 in the CSE treated group (Figure 4), indicating that CSE increased the CSH of *C. albicans*.

### 2.5. CSE Increased the Adhesion Force of C. albicans Biofilms

The adhesion force of *C. albicans* biofilms were calculated from force–distance curves measured by Atomic force microscopy (AFM). As expected, CSE increased the adhesion force of *C. albicans* biofilms (Figure 5). The adhesion force was found to be 5.546 nN in the control group and 11.718 nN in the CSE treated group (Figure 5A). In the control group, the adhesion force mainly varied from 4 to 8 nN. However, in the CSE treated group, the adhesion force mainly varied from 8 to 12 nN (Figure 5B).

### 2.6. CSE Altered Gene Expression in C. albicans

We measured the expression of genes associated with biofilm formation (*ALS1*, *ALS3*, *HWP1* and *EFG1*) by RT-qPCR. The data showed that the expression level of *EFG1* was upregulated, while the expression level changes of genes *ALS1*, *ALS3* and *HWP1* were not statistically significant (Figure 6).

### 2.7. The pH Values of CSE and Boiled CSE

To exclude the effects of pH on the morphogenesis and biofilm formation of *C. albicans*, we examined the pH of CSE, boiled CSE, CSE + YNBB and boiled CSE + YNBB. The pH values of CSE and boiled CSE decreased significantly. However, when mixed with YNBB no significant differences in pH were observed among CSE, boiled CSE and control (Figure 7).

## 3. Discussion

The microbiology analysis revealed that *E. faecalis* and *C. albicans* were highly associated with the persistence of root canal infections [21]. Therefore, the interaction between these two species has been questioned. A previous study found that *E. faecalis* attenuated the hyphal morphogenesis and biofilm formation of *C. albicans* by bacteriocin EntV [17]. However, the interactions between *C. albicans* and bacteria are complex, involving direct cell to cell contact, secretion of small molecules, the use of metabolic by-products, changes in the host environment or a combination of these [22]. In the present study, we found that *E. faecalis* promoted the hypha and biofilm formation of *C. albicans*. Meanwhile, we found that the molecule which mediated the interaction was secreted to extracellular space, since CSE promoted the biofilm formation of *C. albicans* (Figure 1A), but the bacterial cells and intracellular content of *E. faecalis* did not exhibit this effect (Appendix A). From the perspective of this study, it is not clear if *E. faecalis* ATCC 19433 produced EntV or not. We hypothesize that strain-specific molecules produced by different strains of *E. faecalis* could exhibit different effects to *C. albicans*, which need to be verified in future studies.

*E. faecalis* can produce lactic acid in BHI medium and lowers the pH [23]. The growth and morphogenesis of *C. albicans* can be influenced by a change in pH [24,25]. In the present study, CSE and boiled CSE were mixed with equal volumes of YNBB medium to treat *C. albicans*. Therefore, the pH values of CSE and boiled CSE mixed with 50% of YNBB were measured. As shown in Figure 7, CSE and boiled CSE exhibited obviously lower pH values. However, when mixed with equal volumes of YNBB, there were no significant differences in pH among CSE, boiled CSE and control. Therefore, the hypha and biofilm formation of *C. albicans* were not influenced by the pH change of CSE. Moreover, the concentration of lactic acid itself also needed to be considered. We also measured the effect of the supernatant of another two lactic acid bacteria, *Streptococcus gordonii* (CSSg) and *Streptococcus salivarius* (CSSs), on *C. albicans* biofilm formation [26,27]. We verified that *S. gordonii* and *S. salivarius* produced acid in their supernatants, and the pH change could be buffered by YNBB (Appendix A). As shown in Appendix A, CSSg and CSSs exhibited no obvious effect on *C. albicans* biofilm. Therefore, the presence of lactic acid in CSE might be unrelated with the promotion of the hyphal morphogenesis and biofilm formation of *C. albicans.* However, there was a limitation in the study that no direct evidence supported the presence of lactic acid in CSSg and CSSs, even though the pH decrease was observed. Thus, we should apply caution to this result. Meanwhile, we found that the boiled supernatant also clearly promoted biofilm formation of *C. albicans*, indicating that the effective substance is heat stable and unlikely to be a protein, which will be further studied by our group.

*C. albicans* mature biofilms consist of yeast, hyphae and EPS [11]. To explore if CSE increased the hyphal morphogenesis or cell quantity in *C. albicans* biofilms, CFU counting, SEM and CLSM assays were conducted. Unsurprisingly, both hyphal morphogenesis and cell quantity were increased. Hyphae are important in *C. albicans* biofilm development and maintenance, since they contribute to the adhesion and architectural stability of the biofilm [10]. Moreover, hyphal morphogenesis of *C. albicans* contributes to its invasion into oral mucosa [28]. Therefore, the interaction between *E. faecalis* and *C. albicans* might lead to increased microbial invasion into oral mucosa, which was demonstrated in a recent study using an organotypic oral epithelial model [18]. Additionally, EPS also plays an important role in *C. albicans* biofilms. In the mature biofilms, EPS acts as an adhesive glue to connect the overall biofilm structure together [11]. Our data also showed that CSE increased the EPS production of *C. albicans*, which might be one of the mechanisms that *E. faecalis* activated to increase *C. albicans* biofilm biomass. 

As adhesion is a critical step in *C. albicans* biofilm formation and pathogenicity, we subsequently investigated the adhesion force of *C. albicans* biofilms by AFM. The results showed that the adhesion force in the CSE treated group was obviously increased. We put forward two hypotheses to explain this phenomenon. First, EPS acts as a crucial factor which mediates the adhesion of *C. albicans* [29]. Therefore, the increase of EPS production might contribute to the increase of adhesion force. Second, compared with yeast cells, the adhesion force in the surface of hyphal cells was significantly higher [30]. Therefore, CSE might increase the adhesion force by promotion of hyphal morphogenesis. Moreover, we investigated CSH to assess the adhesion ability. CSH has been generally used as an indicator of *C. albicans* adhesion [31,32,33]. Consistent with the result of the AFM assay, our data indicated that CSE increased the CSH. These results converged to prove that *E. faecalis* promoted the adhesion of *C. albicans*, which was essential in biofilm formation.

To clarify the mechanism of the interaction, RT-qPCR was performed in this study. The data showed that CSE upregulated the expression of gene *EFG1*. Efg1, a transcriptional factor, acts as a regulator in initializing hyphal formation, adherence and biofilm maturation [34,35]. Therefore, the upregulation of gene *EFG1* may contribute to *C. albicans* biofilm biomass increase. In the present study, the RNA was extracted from mature biofilms. Als1, Als3 and Hwp1 are cell wall adhesins, promoting adherence during biofilm formation [35]. Therefore, the gene expression of *ALS1*, *ALS3* and *HWP1* might not increase in the maturation stage of biofilm formation. However, the exact mechanisms by which CSE promoted the hyphal morphogenesis and biofilm formation of *C. albicans* are still unclear and need further study.

## 4. Materials and Methods

### 4.1. Strains and Growth Condition

*E. faecalis* ATCC19433, *C. albicans* SC5314 (ATCC MYA-2876), *S. salivarius* ATCC13419 and *S. gordonii* ATCC10558 used in our study were obtained from State Key Laboratory of Oral Diseases (Sichuan University, China). *E. faecalis*, *S. salivarius* and *S. gordonii* were grown individually in a brain–heart infusion (BHI; Oxoid, Basingstoke, UK) agar medium at 37 °C anaerobically (5% H_2_, 5% CO_2_, and 90% N_2_) for 48 h to form single colony. A single colony was transferred to a 15 mL centrifugal tube containing 5 mL of BHI liquid medium. The bacterial suspension was incubated anaerobically at 37 °C overnight to about 3 × 10^9^ CFU/mL for subsequent experiments.

*C. albicans* was grown in a Yeast Extract Peptone Dextrose (YPD; 1% yeast extract, 2% peptone, and 2% D-glucose) agar medium at 35 °C in an aerobically condition for 48 h to form single colony. A single colony was transferred to a 15 mL centrifugal tube containing 5 mL of YPD liquid medium. The fungal suspension was incubated anaerobically at 37 °C overnight to about 3 × 10^7^ CFU/mL for subsequent experiment. YNBB (0.67% YNB, 75 mM Na_2_HPO_4_-NaH_2_PO_4_, 2.5 mM N-acetylglucosamine, 0.2% casamino acids, and 0.5% sucrose) medium was used for *C. albicans* biofilm formation [36].

### 4.2. Supernatant Collection and pH Measurement

Overnight cultured *E. faecalis*, *S. salivarius* and *S. gordonii* were diluted to 1 × 10^7^ CFU/mL and incubated at 37 °C for another 24 h anaerobically. Subsequently, CSE, CSSg and CSSs were collected by centrifugation at 5000 rpm for 10 min and then filter-sterilized using a 0.22-μm pore size filter. When needed, CSE were boiled for 1 h.

Since the biofilm of C. albicans was grown in a mixed medium (50% BHI + 50% YNBB, 50% CSE + 50% YNBB, and 50% boiled CSE + 50% YNBB), we measured the pH value of CSE and boiled CSE along with, or supplemented with, an equal volume of YNBB. BHI was set as control. Briefly, 4 mL of BHI, CSE and boiled CSE with or without 50% of YNBB was added to centrifuge tubes, and the pH values were measured with a pH meter (Mettler Toledo, Shanghai, China). Each sample was analyzed in triplicate.

### 4.3. Biofilm Formation and Biomass Analysis

Overnight cultured *C. albicans* was diluted to 1 × 10^7^ CFU/mL with YNBB. Then, 0.1 mL of the diluted suspensions was inoculated to 24-well plates containing 0.5 mL CSE + 0.4 mL YNBB, or 0.5 mL boiled CSE + 0.4 mL YNBB in each well. The negative control contained 0.5 mL BHI and 0.4 mL YNBB in each well. The plates were incubated at 37 °C for 24 h anaerobically.

Crystal violet staining was performed to investigate the biofilm formation according to a previous study, with some modifications [37]. In brief, the supernatant in each well was removed. Then, the biofilms were gently rinsed three times with sterile phosphate buffer saline (PBS) and dried in the air. To fix the biofilms, 95% methanol was added. After 15 min, the methanol was removed and the biofilms were washed with PBS and dried in the air again. Subsequently, the biofilms were stained with 0.1% (*w/v*) crystal violet for 15 min, followed by removing the crystal violet solution, washing the biofilms with PBS and air drying. The images of biofilms stained by crystal violet were obtained by a stereomicroscope (Leica, Wetzlar, Germany). Then, 100% ethanol was added to each well to extract the crystal violet in biofilms. After 15 min, the ethanol was transferred to another 24-well plate and the optical density at 595 nm was detected using a spectrophotometer (Thermo, Waltham, MA, USA).

### 4.4. CFU Counting

*C. albicans* was grown as described above. After incubation, the supernatant was removed and the biofilms were gently rinsed with sterile PBS. Then the biofilms were scraped and separated by sonication in PBS buffer. Subsequently, the suspension was serially diluted to 10^2^-fold to 10^4^-fold in PBS buffer. A total volume of 30 μL of each diluent was then inoculated to YPD agar plates and incubated aerobically at 35 °C for 48 h. The plates with a CFU count of 30 to 300 were selected for CFU counting.

### 4.5. Scanning Electron Microscopy (SEM)

*C. albicans* was grown as described above, and glass slides were added to the bottom of plates prior to inoculation. A previously described protocol of SEM assay was applied, with some modifications [38]. Briefly, the supernatant was removed and biofilms were gently washed with PBS. Then, the biofilms were fixed with 1 mL of 2.5% glutaraldehyde (*v/v*) under 4 °C for 4 h. Then, the glutaraldehyde was removed and the biofilms were washed with PBS, followed by dehydration with a graded ethanol solutions series (30%, 50%, 70%, 80%, 85%, 90%, 95%, 100%, *v/v*) for 15 min. The glass slides covered with biofilms were stored in 100% ethanol until subsequent analysis. Finally, the samples were taken out and coated with gold and scanned using a scanning electron microscope (Inspect F, FEI, Eindhoven, The Netherlands).

### 4.6. Confocal Laser Scanning Microscopy

*C. albicans* was grown as described above, and glass slides were added to the bottom of plates prior to inoculation. After incubation, the supernatant was removed and the biofilms were washed with 0.9% NaCl. In order to observe the morphology of *C. albicans*, SYTO-9 green fluorescent nucleic acid (excitation 485 nm/emission 498 nm; Invitrogen, Carlsbad, CA, USA) was used to stain the cells at room temperature for 20 min [39]. In order to investigate the biomass of *C. albicans* and its EPS, SYTO 9 nucleic acid was added to stain the cells at room temperature for 20 min, followed by the addition of EPS dye Alexa Fluor 647 (excitation 650 nm/emission 668 nm; Invitrogen, Waltham, MA, USA) to label the EPS at room temperature for 30 min [40]. Finally, the excess stains were removed and the glass slides covered with biofilms were taken out for subsequent analysis. The samples were observed under a confocal laser scanning microscope (CLSM; Olympus, FV3000, Tokyo, Japan) with 60× oil immersion objective lens. IMARIS 9.6.0 software was used for three-dimensional reconstruction. The quantification of EPS/cell ratio was analyzed using Image J COMSTAT software (NIH, Bethesda, MD, USA), according to fluorescence intensity. At least three different positions of each sample were randomly selected for observation. The whole staining procedure was performed in the dark.

### 4.7. Cellular Surface Hydrophobicity (CSH) Assay

CSH was measured by a water–hydrocarbon two-phase assay according to a previous study [32]. *C. albicans* was grown as described above. After incubation, the biofilms were collected and separated by sonication in PBS buffer. Subsequently, the fungal cells were harvested by centrifugation at 4000× *g* for 10 min at 4 °C and re-suspended in YPD medium. The OD_600nm_ was adjusted to 1.0 with YPD medium. Then, 1.2 mL of the fungal suspension was added into an Eppendorf tube. Subsequently, 0.3 mL of octane was added to the top of each tube. The tubes were vortexed for 180 s to intensively mix octane with fungal suspension. Then the tubes were left to stand for another 180 s to separate the aqueous phase and hydrocarbon. Then, the OD_600nm_ of the aqueous phase was measured. For each group, the OD_600nm_ for the group without the supplement with octane was set as inner control. At least three independent experiments were performed. Relative hydrophobicity was obtained from the formula: (OD_600nm_ of inner control–OD_600nm_ mixed with octane)/ OD_600nm_ of inner control.

### 4.8. Atomic Force Microscopy (AFM)

AFM was performed to determine the adhesion force of *C. albicans* biofilms as described earlier with some modifications [30,41]. *C. albicans* was grown as described above, and glass slides were added to the bottom of plates prior to inoculation. After incubation, the supernatant was removed and the biofilms were washed with PBS. The glass slides covered with biofilms were taken out for AFM analysis. The surface adhesion force of *C. albicans* biofilms was measured using a SHIMADZU STM9700 system (Shimadzu, Kyoto, Japan) with tipless AFM probes (0.05 N/m). The adhesion force was examined under a contact model. For each group, 10 different positions were randomly selected and at least 10 force–distance curves at each position were collected.

### 4.9. Real-Time Quantitative PCR (RT-qPCR)

To investigate the gene expression in *C. albicans* biofilms, an RT-qPCR assay was performed according to a previous study [42]. In brief, *C. albicans* was grown as described above. After incubation, the supernatant was removed and the biofilms were immersed in PBS. Then, the biofilms were scraped and collected into Eppendorf tubes. Cells were harvested by centrifugation at 4000× *g* for 10 min at 4 °C, and the supernatant was discarded. Pellets were washed 3 times with sterile PBS. Then, the fungal cells were lysed using Yeast Processing Reagent (Takara, Japan). RNAiso Plus kit (Takara, Japan) was used for RNA extraction. PrimeScript RT Reagent kit (Takara, Japan) was used for cDNA synthesis. The cDNA and RNA qualities were evaluated with a NanoDrop 2000 spectrophotometer (Thermo, Waltham, MA, USA). The cDNA was stored at −20 °C until real-time quantitative PCR (RT-qPCR) analysis. *C. albicans* gene *ACT1* was set as internal control [43]. The primers used in this study are listed in Table 1. The RT-qPCR was carried out with the TB Green Premix Ex Taq II Kit (Takara, Japan) in a LC480 system (Roche, Basel, Switzerland). The cycling procedure started with an initial incubation at 95 °C for 30 s, followed by 40 cycles of 5 s denaturation at 95 °C, 30 s annealing at 55 °C and 30 s extension at 72 °C. The relative transcript level of the studied genes was determined with the 2^−ΔΔCT^ method. For each sample, three repeats were performed.

### 4.10. Data Analysis

Each experiment was independently repeated at least three times. The data were analyzed with GraphPad Prism 8.0 (GraphPad, La Jolla, CA, USA). One-way analysis of variance (ANOVA) was used to analyze the result of crystal violet staining, CFU and RT-qPCR assay, and post hoc test was applied for multiple testing. Student *t* test was used to analyze the result of the CLSM, AFM and CSH assays. The difference was considered as statistically significant when *p* < 0.05.

## 5. Conclusions

*E. faecalis* has traditionally been considered to suppress the hyphal morphogenesis and biofilm formation of *C. albicans* by its secreted bacteriocin EntV. Recently, *E. faecalis* was found to enhance *C. albicans* invasion in an organotypic oral epithelial model. In this study we found that the supernatant of *E. faecalis* enhanced the hyphal morphogenesis and biofilm formation of *C. albicans in vitro*. EPS, CSH, adhesion force and the expression of genes related with biofilm formation were all increased. Therefore, the current study suggests that *E. faecalis* is able to promote the hyphal morphogenesis and biofilm formation of *C. albicans*. Nevertheless, further studies are needed to clarify the molecular mechanism of the promotion effect.

## Figures and Tables

**Figure 1 pathogens-11-01177-f001:**
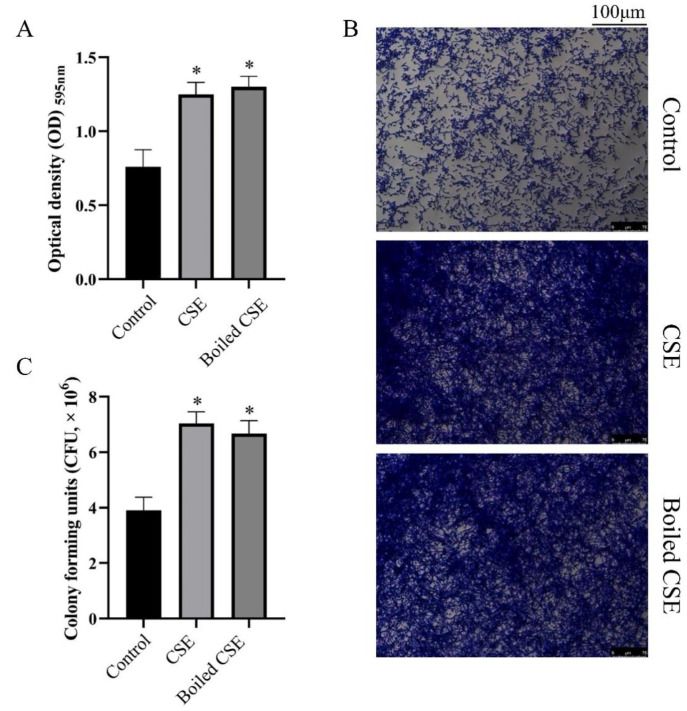
The effect of the culture supernatant of *E. faecalis* (CSE) on biofilm formation of *C. albicans*. (**A**) Biofilm biomass under the effect of CSE and boiled CSE (means ± SD) measured by crystal violet staining. (**B**) Crystal violet stained biofilms obtained by a stereomicroscope. (**C**) The mean CFU counts of *C. albicans* biofilms in one well (means ± SD). * *p* < 0.05. Control: 50% YNBB + 50% BHI. CSE: 50% YNBB + 50% CSE. Boiled CSE: 50% YNBB + 50% boiled CSE. *C. albicans* was grown in CSE or boiled CSE for 24 h.

**Figure 2 pathogens-11-01177-f002:**
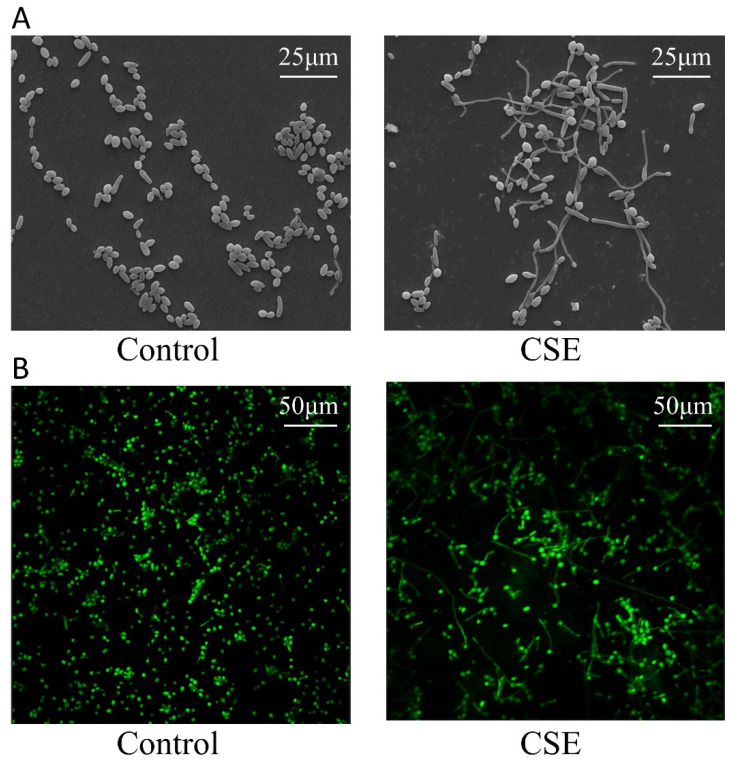
The effect of the culture supernatant of *E. faecalis* (CSE) on *C. albicans* hyphal morphogenesis. (**A**) Scanning electronic microscope (SEM) images of *C. albicans.* (**B**) Confocal laser scanning microscope (CLSM) images of *C. albicans.* Control: 50% YNBB + 50% BHI. CSE: 50% YNBB + 50% CSE. *C. albicans* was grown in CSE for 24 h.

**Figure 3 pathogens-11-01177-f003:**
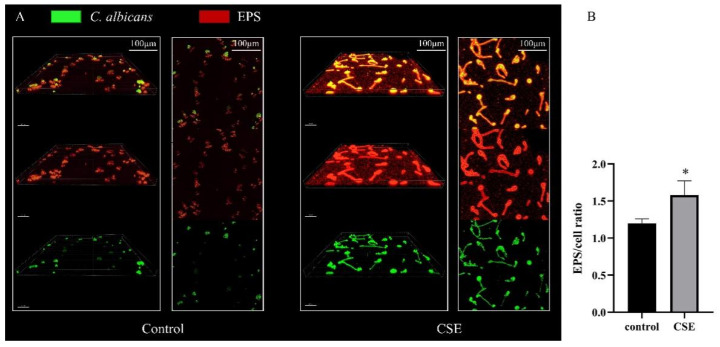
The effect of the culture supernatant of *E. faecalis* (CSE) on *C. albicans* EPS production measured by CLSM assay. (**A**) *C. albicans* biofilms stained with SYTO 9 (green, fungi) and Alexa Fluor 647 (red, EPS). Images were obtained at 60× magnifications. IMARIS 9.6.0 was used for 3-D reconstruction. (**B**) Quantitative analysis of EPS/cell ratio by Image J COMSTAT (means ± SD) according to fluorescence intensity. * *p* < 0.05. Control: 50% YNBB + 50% BHI. CSE: 50% YNBB + 50% CSE. *C. albicans* was grown in CSE for 24 h.

**Figure 4 pathogens-11-01177-f004:**
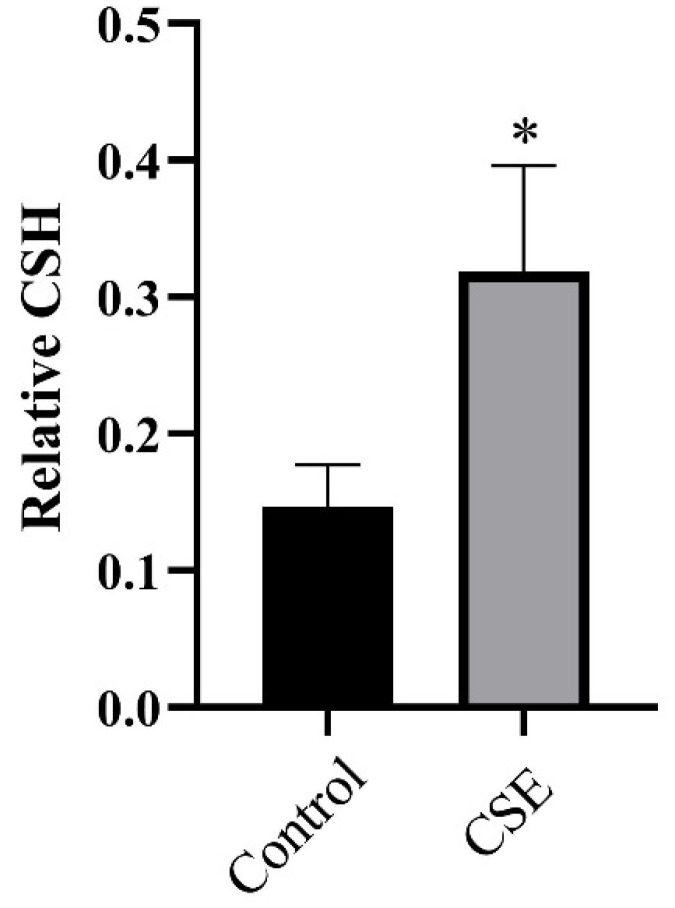
The effect of the culture supernatant of *E. faecalis* (CSE) on cellular surface hydrophobicity (CSH) of *C. albicans.* A water–hydrocarbon two-phase assay was performed to evaluate CSH (means ± SD). * *p* < 0.05. Control: 50% YNBB + 50% BHI. CSE: 50% YNBB + 50% CSE. *C. albicans* was grown in CSE for 24 h.

**Figure 5 pathogens-11-01177-f005:**
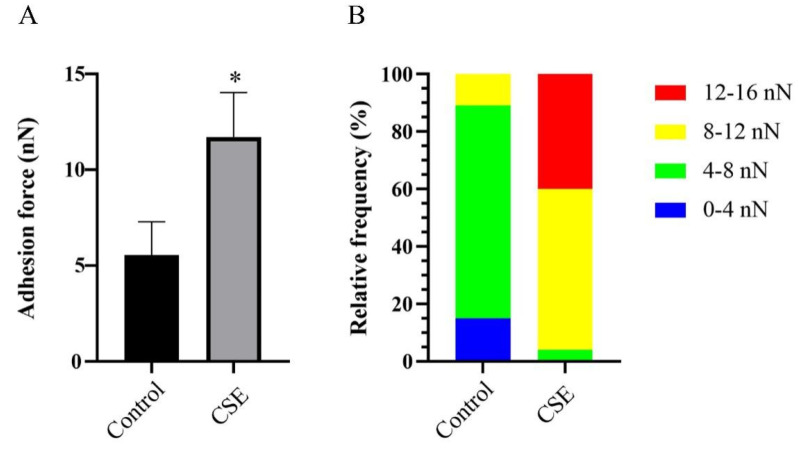
The effect of the culture supernatant of *E. faecalis* (CSE) on the adhesion force of *C. albicans* biofilms by AFM assay. (**A**) Data of the adhesion force measured by AFM (means ± SD). * *p* < 0.05. (**B**) The distribution of the adhesion force based on 100 force–distance curves at 10 random positions of each group. Control: 50% YNBB + 50% BHI. CSE: 50% YNBB + 50% CSE. *C. albicans* was grown in CSE for 24 h.

**Figure 6 pathogens-11-01177-f006:**
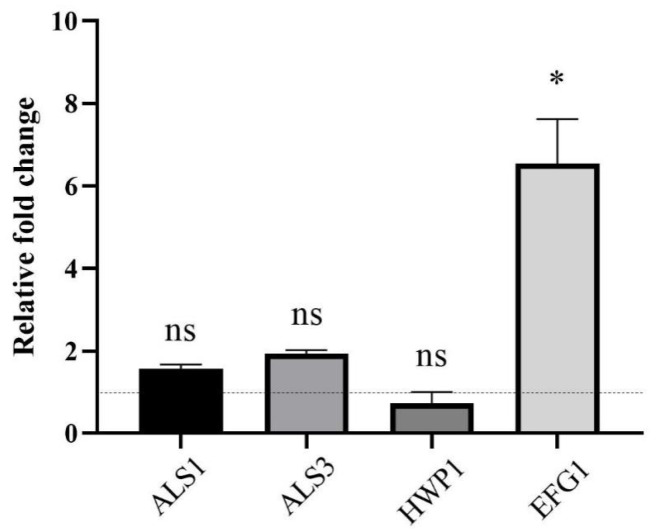
The effect of the culture supernatant of *E. faecalis* (CSE) on the gene expression of *C. albicans* determined by real-time quantitative PCR. Gene *ACT1* was used as inner control. The values are represented as means ± SD from three independent experiments. * *p* < 0.05, ns: no significance. *C. albicans* was grown in CSE for 24 h.

**Figure 7 pathogens-11-01177-f007:**
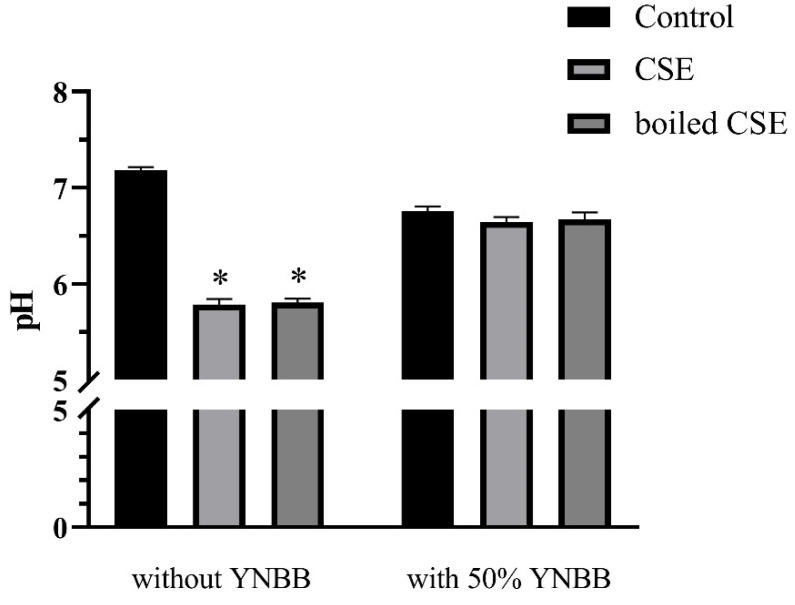
The pH values of CSE and boiled CSE with or without 50% YNBB. BHI liquid medium was set as control. The values are shown as means ± SD from three independent experiments. * *p* < 0.05.

**Table 1 pathogens-11-01177-t001:** *C. albicans* primers sequence used in this study for real-time quantitative PCR (RT-qPCR) analysis.

Gene	Forward Primer	Reverse Primer
ACT1	GCGGTAGAGAGACTTGACCAACC	GACAATTTCTCTTTCAGCACTAGTAGTG
ALS1	CCAAGTGTTCCAACAACTGAA	GAACCGGTTGTTGCTATGGT
ALS3	CTAATGCTGCTACGTATAATT	CCTGAAATTGACATGTAGCA
HWP1	TGGTGCTATTACTATTCCGG	CAATAATAGCAGCACCGAAG
EFG1	TATGCCCCAGCAAACAACTG	TTGTTGTCCTGCTGTCTGTC

## Data Availability

All data used to support this study are available from the corresponding author on reasonable request.

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
