# Peer review of "Culture Supernatant of Enterococcus faecalis Promotes the Hyphal Morphogenesis and Biofilm Formation of Candida albicans"

_pathogens, 2022, doi:10.3390/pathogens11101177_

Round 1

Reviewer 1 Report (New Reviewer)

Summary:

Enterococcus faecalis and Candida albicans occupy overlapping niches and both can be isolated from infected canals. Studying the interactions between these two bacterium and fungus could provide insights on their coexistence in humans during both healthy and disease status. The authors applied various approaches to demonstrate that the culture supernatant of Enterococcus faecalis promotes hyphal morphogenesis and biofilm formation of Candida albicans. Congruously, exopolymeric substances, cell surface hydrophobicity, adhesion force and biofilm-related gene EFG1 were increased in Candida albicans upon the treatment with culture supernatant of Enterococcus faecalis. The authors here presented a very interesting phenotype, however, neither of the experiment or text contents provided is informative enough to advance the knowledge of the current topic. Additional experiments and/or discussions are advised.

Major points:

1.     The author only provided one pair of example here with E. faecalis ATCC19433 and C. albicans SC5314, which may not be generalizable in other conditions. Additional strains, especially recent clinical isolates would be very interesting to look at.

2.     Relating to the previous point. The authors just mentioned that there were controversial results of whether E. faecalis promote/inhibit C. albicans hyphal morphogenesis. This is a great discussion point. Does E. faecalis ATCC19433 produce entV under the experimental condition? Would C. albicans SC5314 response to entV if it is produced? Why you saw the opposite? The authors just stated “the cross-kingdom interaction between these two species is more complicated than what we understand” without providing an explanation or a hypothesis. Additional experiments and discussion to address this controversy would be very informative.

3.     It is better to see the authors make additional efforts to try to identify the potential molecules that promote hyphal morphogenesis and biofilm formation.

4.     Line 143-158 belongs to the results section. It is an interesting piece of data to show that pH is not driving the hyphal morphogenesis and biofilm formation of Candida albicans. These could be incorporated as a main figure.

5.     Evidence needed here to show that lactic acid is produced by CSSg and CSSs to exclude the potential impacts of lactic acid.

6.     What are the genes of ALS1, ALS3, HWP1. How to explain only EFG1 increased but not others?

Minor points:

1.     Line 36-37, “In vitro studies demonstrated that it could penetrate into the dentin tubules up to 100 μm, and can be found after 12 months, making the elimination difficult.” The authors did specify “it” or “after 12 months”. Please clarify what these phrases are referring to.

2.     Line 50-52, are these two E.faecalis strains the same? What could explain the controversy?

3.     Line 88, EPS production?

4.     Please specify how you determine the EPS/cell ratio?

5.     Line 115, (Fig. 5B)

6.     Line 199, 35 degree?

7.     In material methods, “C. albicans at 1×106 CFU/ml were inoculated to 24-well plates containing 0.5 ml CSE + 0.5ml YNBB, or 0.5 ml boiled CSE + 0.5ml YNBB in each well. The negative control contained 0.5 ml BHI and 0.5 ml YNBB in each well. The biofilms of C. albicans were incubated at 37℃ _for 24h in an anaerobic condition (5% H2, 5% CO2, and 90% N2).” was repeated in almost every paragraph. Maybe just wright it once and refer to if later on as “C. albicans was grown as described above”

8.     Line 254 “At least three random points of each sample were selected for observation.” What do the random points mean here?

Author Response

Major points:

  1. The author only provided one pair of example here with E. faecalis ATCC19433 and C. albicans SC5314, which may not be generalizable in other conditions. Additional strains, especially recent clinical isolates would be very interesting to look at.

Reply to comment 1: Thanks for your comment. It is sure that introduction of more strains would benefit to the robustness of the study. In this study, we studied E. faecalis ATCC 19433 (also named as NCTC775 and NCDO 5681) and C. albicans SC5314, because they are the standard strains of E. faecalis and C. albicans species, respectively.

  1. Relating to the previous point. The authors just mentioned that there were controversial results of whether E. faecalis promote/inhibit C. albicans hyphal morphogenesis. This is a great discussion point. Does E. faecalis ATCC19433 produce entV under the experimental condition? Would C. albicans SC5314 Reply to entV if it is produced? Why you saw the opposite? The authors just stated “the cross-kingdom interaction between these two species is more complicated than what we understand” without providing an explanation or a hypothesis. Additional experiments and discussion to address this controversy would be very informative.

Reply to comment 2: Thanks for your nice comment. We did not detect EntV because it was not the key point in our study. No study supported that E. faecalis ATCC 19433, the strain used in the present study, could produce EntV. Therefore, we are not sure whether EntV is present in CSE or not. C. albicans SC5314 is able to respond to EntV according to a previous study [17]. We have added “From the perspective of this study, it is not clear if E. faecalis ATCC 19433 produced EntV or not. We hypothesize that strain-specific molecules produced by different strains of E. faecalis could exhibit different effects to C. albicans, which need to be verified in future studies.” to the first paragraph of Discussion.

  1. It is better to see the authors make additional efforts to try to identify the potential molecules that promote hyphal morphogenesis and biofilm formation.

Reply to comment 3: Thanks for your comment. The molecular mechanism that CSE promote the hyphal morphogenesis and biofilm formation of C. albicans is attractive. We will do our best to identify the potential molecules.

  1. Line 143-158 belongs to the results section. It is an interesting piece of data to show that pH is not driving the hyphal morphogenesis and biofilm formation ofCandida albicans. These could be incorporated as a main figure.

Reply to comment 4: Thanks for your nice comment. The content of line 143-158 has been put into the Results section. We added “2.7. The pH values of CSE and boiled CSE” and “2.8. The culture supernatant of Streptococcus gordonii (CSSg) and Streptococcus salivarius (CSSs) exhibited no effect on C. albicans biofilms” in results section.

The corresponding methods were also added to the Materials and Methods section.

In 4.1. Strains and growth condition, the first paragraph was modified to “E. faecalis ATCC19433, C. albicans SC5314 (ATCC MYA-2876), S. salivarius ATCC13419 and S. gordonii ATCC10558 used in this study were provided by State Key Laboratory of Oral Diseases (Sichuan University, China). E. faecalis, S. salivarius and S. gordonii were grown individually in a Brain-heart infusion (BHI; Oxoid, Basingstoke, UK) agar medium at 37℃ in an anaerobic condition (5% H2, 5% CO2, and 90% N2) for 48 h to form single colony. A single colony was transferred to a 15 ml centrifugal tube containing 5 ml of BHI liquid medium. The bacterial suspension was incubated anaerobically at 37℃ overnight to about 3×109 CFU/ml for subsequent experiment.”

In 4.2. Supernatant collection and pH measurement, we added the second paragraph “Since the biofilm of C. albicans was grown in a mixed medium (50% BHI + 50% YNBB, 50% CSE + 50% YNBB, and 50% boiled CSE + 50% YNBB), we measured the pH value of CSE and boiled CSE along or supplemented with equal volume of YNBB. BHI was set as control. Briefly, 4ml of BHI, CSE and boiled CSE with or without 50% of YNBB was added to centrifuge tubes, and the pH values were measured with a pH meter (Mettler Toledo, China). Each sample was analyzed in triplicate.”

In 4.3. Biofilm formation and biomass analysis, we added “To exclude the effect of lactic acid in CSE, the effects of the culture supernatant of another two lactic acid bacteria, S. salivarius and S. gordonii, were also measured. Briefly, 0.1 ml of the diluted suspensions of C. albican was inoculated to 24-well plates containing 0.5 ml CSSs + 0.4ml YNBB, or 0.5 ml CSSg + 0.4ml YNBB in each well. The negative control contained 0.5 ml BHI and 0.4 ml YNBB in each well. The plates were incubated at 37℃ for 24h in an anaerobic condition (5% H2, 5% CO2, and 90% N2)” to the second paragraph.

  1. Evidence needed here to show that lactic acid is produced by CSSg and CSSs to exclude the potential impacts of lactic acid.

Reply to comment 5: Thanks for your comment. We have added two references (reference 21 and 22) to section 2.8 “To exclude the effects of lactic acid on the biofilm formation of C. albicans, we incorporated another two lactic acid producing bacteria, S. gordonii and S. salivarius[21,22]”, the second paragraph of section 3 “We also measured the effect of the supernatant of another two lactic acid bacteria, Streptococcus gordonii (CSSg) and Streptococcus salivarius (CSSs), on C. albicans biofilm formation[21,22]”, and the second paragraph of section 4.3 “To exclude the effect of lactic acid in CSE, the effects of the culture supernatant of another two lactic acid bacteria, S. salivarius and S. gordonii, were also measured[21,22]”. These two study verified the production of lactic acid by S. salivarius and S. gordonii.

  1. What are the genes of ALS1, ALS3, HWP1. How to explain only EFG1 increased but not others?

Reply to comment 6: Thanks for your comment. ALS1, ALS3, HWP1 are cell wall adhesin. EFG1 is a transcriptional factor which participates in the adherence and maturation of C. albicans biofilm formation. ALS1, ALS3, HWP1 promote the adherence of C. albicans but not maturation[35]. In our study, mature biofilms were analysed. Therefore, we speculate that the gene expression of ALS1, ALS3, HWP1 might increase in adherence stage and return to control level on maturation stage. We have modified some expression in the last paragraph of Discussion to “Efg1, a transcriptional factor, acts as a regulator in initialing hyphal formation, adherence and biofilm maturation[34,35]. Therefore, the upregulation of gene EFG1 may contribute to C. albicans biofilm biomass increase. In the present study, the RNA was extracted from mature biofilms. Als1, Als3 and Hwp1 are cell wall adhesin, promoting the adherence during biofilm formation[35]. Therefore, the gene expression of ALS1, ALS3 and HWP1 might not increase in the maturation stage of biofilm formation.”

Minor points:

  1. Line 36-37, “In vitro studies demonstrated that it could penetrate into the dentin tubules up to 100 μm, and can be found after 12 months, making the elimination difficult.” The authors did specify “it” or “after 12 months”. Please clarify what these phrases are referring to.

Reply to comment 1: Thanks for your nice comment. We have modified this sentence to “In vitro studies demonstrated that E. faecalis could penetrate into the dentin tubules up to 100 μm, and E. faecalis inoculated into root canals could be revived after 12 months, making the elimination difficult”

  1. Line 50-52, are these twoE.faecalis strains the same? What could explain the controversy?

Reply to comment 2: Thanks for your comment. These two E.faecalis strains are the same[17,18]. Maybe the difference of experimental conditions made the difference of results.

  1. Line 88, EPS production?

Reply to comment 3: Thanks for your comment. We have replaced “production” with “EPS production” in line 88.

  1. Please specify how you determine the EPS/cell ratio?

Reply to comment 4: Thanks for your comment. The EPS/cell ratio was determined according to fluorescence intensity. We have modified the legend of Figure 3 (B) to “Quantitative analysis of EPS/cell ratio by Image J COMSTAT (means ± SD) according to fluorescence intensity”. The corresponding description in section 4.6 was modified to “And the quantification of EPS/cell ratio was analyzed using Image J COMSTAT software (NIH, USA) according to fluorescence intensity.”

  1. Line 115, (Fig. 5B)

Reply to comment 5: Thanks for your comment. We have added “(Fig. 5B)” to the end of the sentence “In the control group, the adhesion force mainly varies between 4 and 8 nN. However, in the CSE treated group, the adhesion force mainly varies between 8 and 12 nN (Fig. 5B).”

  1. Line 199, 35 degree?

Reply to comment 6: Thanks for your comment. C. albicans was revived at 35 ℃

  1. In material methods, “C. albicans at 1×106 CFU/ml were inoculated to 24-well plates containing 0.5 ml CSE + 0.5ml YNBB, or 0.5 ml boiled CSE + 0.5ml YNBB in each well. The negative control contained 0.5 ml BHI and 0.5 ml YNBB in each well. The biofilms of C. albicans were incubated at 37℃_for 24h in an anaerobic condition (5% H2, 5% CO2, and 90% N2).” was repeated in almost every paragraph. Maybe just wright it once and refer to if later on as “C. albicans was grown as described above”

Reply to comment 7: Thanks for your nice comment. We have replaced the repeated sentences with “C. albicans was grown as described above” from section 4.4 to section 4.9.

  1. Line 254 “At least three random points of each sample were selected for observation.” What do the random points mean here?

Reply to comment 8: Thanks for your nice comment. Random points here mean different positions which are randomly selected on one sample. We have modified this sentence to “At least three different positions of each sample were randomly selected for observation.”

Reviewer 2 Report (New Reviewer)

The authors have done a meticulous study to show the cross-kingdom interactions between dental pathogens.

Author Response

Comments:The authors have done a meticulous study to show the cross-kingdom interactions between dental pathogens.

Reply: Thank you very much. Your comments provide many useful ideas for us to modify the manuscript.

Reviewer 3 Report (Previous Reviewer 2)

The authors have, as far as I can tell, modified the suggested improvements. I have no more additional comments.

Author Response

Comments: The authors have, as far as I can tell, modified the suggested improvements. I have no more additional comments.

Reply: Thank you very much. Your meticulous comments help us improve the study and manuscript.

Round 2

Reviewer 1 Report (New Reviewer)

The authors have addressed most of my questions and concerns. This is a much-improved manuscripts thanks to their efforts. 

For the CSSg and CSSs results, citing previous papers does not serve as evidence that these bacteria produced lactic acid in your condition. Did you measure the pH of their supernatant?  Consider putting figure 2.8 back to supplementals and discuss the potential limitations.

Author Response

Dear reviewer:

Thank you very much for giving us a lot of useful advice about this manuscript. We measured the pH of CSSg and CSSs. The pH of CSSg and CSSs was added to supplementary files. As shown in Fig S3, the pH of CSSg and CSSs decreased significantly, and the pH change could be buffered by YNBB. We have put figure 2.8 back to supplementary files. The sentences “However, there was a limitation in the study that no direct evidence supported the presence of lactic acid in CSSg and CSSs, even though the pH decrease was observed. Thus we should take this result with more caution.” were added to the 2nd paragraph of Discussion to state the limitations.

This manuscript is a resubmission of an earlier submission. The following is a list of the peer review reports and author responses from that submission.

Round 1

Reviewer 1 Report

This manuscript reports the effect of culture supernatant of E.faecalis on biofilm formation and yeast-hyphal transition in Candida albicans. The study presented in this manuscript is interesting and concerns a very current issue. However this manuscript needs to be modified to be improved.

My main comments about the manuscript are described below:

  1. Genus names are not italicized line 23
  2. Line 195: a word is missing “CSE and ?”
  3. Line 43-45: “Many bacterial species have been found to exhibit a synergic relationship with C. albicans, including Staphylococus aureus, Streptococcus mutans, Streptococcus…” other references should be added.
  4. Several questions about the supernatant of E.faecalis:

- what is the ph of the supernatant and CSE boiled? it is known that the morphology of C.albicans can be influenced by the pH.

- Of what is the CSE composed? Have the authors analyzed it?

-Please, specify in the text what are the expected differences between the supernatant and the boiled supernatant?

-In the manuscript there are several references to bacteriocin EntV (lines 46,124 and 274) . The reviewer assumes that boiling the supernatant eliminates this bacteriocin but did you prove that EntV was present in the supernatant before boiling?

  1. Fig 1C: why not present the CFU results for cse boiled? please add them
  2. Fig 6 should be moved to the results section
  3. Concerning real time quantitative PCR:

In my opinion, it lacks precisions: how long was the CSE in contact with C. albicans? Was the fungi grown with? on which support? Please also precise the support to biofilm formation for SEM and CSLM

  1. from my point of view it would be necessary to add the processing times with the CSE to the legends of the figures
  2. The format used for the references does not seem to match the recommendations of the journal, please modify

Reviewer 2 Report

This manuscript presents data on the effect of a culture supernatant of one E. faecalis strain on the biofilm formation, hyphal production and some cell-wall associated factors of one C. albicans strain. On the basis of this research a mechanism is proposed.

The data from this study are interesting and important. My main concern about this study is that the authors seem to be too focused on the mechanism and the evidence is not enough to support this. Especially the last paragraph of the discussion is too much speculation.

Furthermore, some controls in their study seem to be lacking, although I think that they have been performed but are not mentioned in the manuscript.

Most important point:

  1. The authors speak about synergism between the two strains, but that is not the case. In synergy there is a mutual benefit and in this study the faecalis does not benefit at all from the growth of Candida; growth of this strain under these conditions has not been tested in this study.
  2. One of the affects shown in this study can be lowering the pH of the CSE because of the glycolytic growth of faecalis. The authors inserted somewhere that they have been using culture supernatant of a fusobacterium strain as control. This has not been shown in the data of the paper itself; only in the supplementary file this is mentioned. Although it is good to have such a control, it not sufficient. E faecalis is an organism that needs carbohydrates and lowers the environmental pH by producing lactate. Fusobacteria are proteolytic organisms, grown anaerobically and do induce a higher pH from the production of NH4+. In this study the authors still cannot rule out an environmental effect of growth; I suggest that another oral Streptococcus strain is used as control, such as S. mitis or S. oralis. The reason that these do not appear in endodontic inflammatory processes regularly is no argument, because the whole study is performed under laboratory conditions and do not at all simulate the endodontic environment.
  3. This is also a major point: the endodontic environment is an anaerobic environment; with an increased CO2 It is known that increased CO2 concentrations induce hyphal growth of yeast. This should be taken into account in this study. Also, the lack of oxygen in the endodontic environment can play an important role because candidal growth is affected by oxygen. The authors should at least discuss this, and they should provide data on these mechanisms. Better would be that they perform their studies under decreased oxygen and increased CO2 conditions.
  4. The authors present data on upregulation of several candida genes involved in processes from which the authors think they may play a role in the proposed mechanisms. Throughout the manuscript, different genes are presented on several places, but this is not done consequently. Not all data of the genes mentioned in the M&M are presented, and in the discussion, some other names appear without explaining why. In the M&M: ACT1, ALS1, ALS3, HWP1, EFG1, ERG6, ERG11, ERG20; Figure 6: ALS1, ALS3, HWP1, EFG1, ERG6, 11 & 20; in the discussion: ALS1, ALS3, EFG1, ERG6, 11 & 20, MDR1 & CDR2. It seems, Furthermore, that only 1 gene (ACT1) has been used as control and with such a large number of genes tested, at least 2 control genes should be included. Also, the statistics should be improved by applying a post hoc test as a correction for multiple testing. In that case, it may be possible that the upregulation of ALS1 and possibly also ALS2 is not statistically significant.
  5. The authors have been using only one strain from each genus. It may be possible that they are studying a strain specific characteristic and should realise this. Introduction of more strains would benefit to the robustness of their study.

I have a few detailed points:

  1. P1, L30: The fact the E. faecalis is associated with secondary endodontal infections does not mean that this bacterium is causing the disease.
  2. P1, L33: It is not true that these bacteria remain viable for 12 months. They can be found still after 12 months, but there is a tremendous turn-over in growing cells and dying cells. Please be more specific.
  3. P5, L128: there is not synergy studied in the present manuscript
  4. In M&M: the authors report a number of CFU/ml used in their experiments. Better is to use optical density, since faecalis can produce long chains, in a strain-dependent manner, and the number of bacterial cells can differ strongly between strains and experimental conditions.
  5. P6, L175 and further: this is much too speculative since no reliable data are presented. Only the upregulation of a few genes, but possible many other genes are also upregulated and possible others downregulated. When you do not look at other things or mechanisms, you will not see them.

Round 2

Reviewer 1 Report

The authors responded to all comments of the reviewer and made changes in the manuscript and figures as requested. It would be interesting to continue the study by analysing CSE, studying more genes in RT-qPCR and studying another strain of E. faecalis
Nevertheless this study brings interesting information . 

Reviewer 2 Report

Most of my comments have been processed correctly. 

I still have a problem, however, with the interpretation of the data. In lines 189 and further, the authors mention two times ALS3 genes; I think they mean ALS 1 and ALS 3. They should, however, realise that "a not statistical significant increase" is actually "no increase". So this paragraph should be rewritten. 

I am, furthermore, also not pleased that the speculation about ergosterol and farnesol still is in the paper. This is speculation that has not been introduced in the introduction. The reason why the authors included to study these genes may be a part of their hypothesis, but this should be explained better in the introduction. But then I wonder why these are the genes that has been studied and no other controls have been included. There may be a dozen of other genes that can be either up- or down-regulated and the authors need to explain better why they have chosen for such a small number of possible mechanisms. They should realise that you will not see what you will not study, so the speculation is for me too much. My advice is still to remove most of these speculations.

The authors corrected for a pH drop by diluting the CSE with a buffered medium. That is fine, when only pH is concerned. But it is also the concentration of lactic acid and that is not controlled for. Furthermore, when introducing a proteolytic bacterium as control, it is a control for just bacterial growth, but not for creating a special ecological environment. The paragraph that is dealing with this issue is much to long and can be shortened. But I still have the opinion that the control, F. nucleatum, is not a correct control. Fn uses proteins for its growth and does not produce lactate. This bacterium is growing as a part of a complex ecosystem, so the reason that "it is also found in endodontic lesions" is not a valid one. The authors should mention this at least in the discussion, but not in such an explicit way as done on line 148 and further. This is much to specific and needs to be included in the M&M section. But my main point of critics  remains: Fn is not a suitable control to compare in vitro growth of Ef. 

The growth conditions in this study are not clearly stated everywhere. Now it has been described in the adhesion experiments that growth took place under anaerobic conditions, but was that also done for the other sub-studies? For example lines 219 and 242.

The authors claim that they have performed post-hoc analyses for multiple testing (which I can see from the graphs), but that is not stated on line 295.
